# Correlation between the Skin Permeation Profile of the Synthetic Sesquiterpene Compounds, Beta-Caryophyllene and Caryophyllene Oxide, and the Antiedematogenic Activity by Topical Application of Nanoemulgels

**DOI:** 10.3390/biom12081102

**Published:** 2022-08-10

**Authors:** Patrícia Weimer, Tainá Kreutz, Renata P. Limberger, Rochele C. Rossi, Ádley A. N. de Lima, Valdir F. Veiga, Bibiana Verlindo de Araújo, Letícia S. Koester

**Affiliations:** 1Programa de Pós-Graduação em Ciências Farmacêuticas, Faculdade de Farmácia, Universidade Federal do Rio Grande do Sul (UFRGS), Porto Alegre 90610-000, RS, Brazil; 2Programa de Pós-Graduação em Nutrição e Alimentos, Universidade do Vale do Rio dos Sinos (UNISINOS), São Leopoldo 93022-000, RS, Brazil; 3Programa de Pós-Graduação em Ciências Farmacêuticas, Departamento de Farmácia, Universidade Federal do Rio Grande do Norte, Natal 59012-570, RN, Brazil; 4Instituto Militar de Engenharia (IME), Rio de Janeiro 22290-270, RJ, Brazil

**Keywords:** copaiba, *Copaifera multijuga*, beta-caryophyllene, caryophyllene oxide, nanoemulgel, anti-inflammatory, HET-CAM assay

## Abstract

Sesquiterpene compounds are applied as permeation promoters in topical formulations. However, studies exploring their impact on nanostructured systems, changes in permeation profile, and consequently, its biological activity are restricted. This study aimed to investigate the correlation between the skin permeation of the major sesquiterpenes, beta-caryophyllene, and caryophyllene oxide from the oleoresin of *Copaifera multijuga*, after delivery into topical nanoemulgels, and the in vivo antiedematogenic activity. First, ten nanoemulgels were prepared and characterized, and their in vitro permeation profile and in vivo anti-inflammatory activity were evaluated. In equivalent concentrations, β-caryophyllene permeation was greater from oleoresin nanoemulgels, resulting in greater in vivo antiedematogenic activity. However, an inverse relationship was observed for caryophyllene oxide, which showed its favored permeation and better in vivo anti-inflammatory effect carried as an isolated compound in the nanoemulgels. These results suggest that the presence of similar compounds may interfere with the permeation profile when comparing the profiles of the compounds alone or when presented in oleoresin. Furthermore, the correlation results between the permeation profile and in vivo antiedematogenic activity corroborate the establishment of beta-caryophyllene as an essential compound for this pharmacological activity of *C. multijuga* oleoresin.

## 1. Introduction

Sesquiterpenes are volatile compounds produced by plant species from the condensation of isoprene to dimethyl diphosphate. They have a carbon chain formed by 15 carbon atoms, which can be organized in acyclic or cyclic structures and can also have additional organic functions or oxygen atoms (oxygenated sesquiterpenes). Among the pharmacological activities performed by these compounds are antimicrobial, expectorant, and anti-inflammatory activities [1].

The sesquiterpene β-caryophyllene (CAR, Figure 1A) has been experimentally shown to contribute to the anti-inflammatory activity of some medicinal plants, especially of the genus *Copaifera*. Recent studies have described the mechanisms of CAR action in inflammatory processes, in particular, the action of CAR on the enzymes, cyclooxygenase-2 (COX-2), and lipoxygenase-5 (LOX-5), thereby reducing the release of inflammatory mediators, mitigating the symptoms of pain and edema, and highlighting its healing property [2,3,4,5,6]. The broad pharmacological potential of CAR is highlighted in the reviews by Sharma et al. (2016), Francomano et al. (2019), and Rahimi and Askari [7,8,9]. Additionally, studies have highlighted CAR as a promising compound for pain management owing to its action in the endocannabinoid system via selective interaction with cannabinoid 2 (CB2) receptors, thus providing analgesic action [2,10,11].

In the *Copaifera multijuga* oleoresin, in addition to CAR, a high concentration of caryophyllene oxide (OX, Figure 1B) could also be found [12,13]. OX corresponds to an oxygenated derivative of CAR, which is present in the oleoresin at concentrations close to those of CAR. Despite the large presence of OX, studies have not yet investigated the contribution of OX to the anti-inflammatory activity of the *C. multijuga* oleoresin. In this context, studies that evaluate in parallel the importance of the oxygenated derivatives of caryophyllene in the pharmacological activity of *C. multijuga* oleoresin are necessary. Furthermore, studies that effectively compare the phytocomplex with isolated compounds at equivalent concentrations are also needed to establish the correlations of the permeation profile with the pharmacological activity. Such studies are especially important considering the traditional topical use of this oleoresin and the therapeutic potential of topical administration.

Nonetheless, the topical administration of these compounds or of the oleoresin in its free form, i.e., not loaded into pharmaceutical delivery systems, experience significant limitations due to lipophilicity, an unpleasant unctuous aspect, and low skin permeability [14,15,16,17]. To overcome this problem, technological approaches can be employed in order to increase patient compliance and skin permeation, such as nanoemulgels, which are generated through the addition of a gelling polymer to nanoemulsions. By carrying volatile lipophilic compounds in topical nanoemulgels, many advantages are gained, such as reducing losses due to volatilization or photodegradation, potentializing therapeutic activity, promoting skin permeation, and increasing acceptability by the patient, compared to the use of nanoemulsions [15,18,19].

The aim of this study was to investigate the correlation between the in vitro skin permeation profiles of the major sesquiterpenes from the oleoresin of *C. multijuga*, beta-caryophyllene, and caryophyllene oxide after delivery into topical nanoemulgels, and their impact on antiedematogenic activity in an in vivo ear edema model. The technological characterization of the formulations and in vitro safety profile assessment was also conducted.

## 2. Materials and Methods

### 2.1. Chemicals and Reagents

CAR (91.00%), OX (96.90%), sorbitan monooleate (Span^®^ 80), polyoxyethylenesorbitan monolaurate (Tween^®^ 20), arachidonic acid (99.00%), and 3,3′,5,5′-tetramethylbenzidine dihydrochloride hydrate were purchased from Sigma (St. Louis, MO, USA). Medium-chain triglycerides (MCT: capric and caprylic acids) were purchased from Mapric Greentech Company (Diadema, SP, Brazil). Indomethacin (96.00%) and hydroxyethyl cellulose (HEC) were purchased from Henrifarma Produtos Químicos e Farmacêuticos (São Paulo, SP, Brazil). All other chemicals or reagents were of analytical grade. Ultrapure water was obtained from a Milli-Q apparatus (Millipore, Merck, Darmstadt, Germany).

### 2.2. Plant Material

*C. multijuga* oleoresin (COP) was collected from the Reserva Florestal Ducke, Instituto Nacional de Pesquisas da Amazônia (INPA), Manaus (2°57′43″ S, 59°55′38″ W), Brazil, and was identified by the botanist José Edimilson da Costa Souza. The voucher specimen was deposited at the INPA Herbarium, and the plant name was checked in The Plant List database and International Plant Names Index. Previously, the research with genetic material from Brazilian biodiversity was registered in the Sistema Nacional de Gestão do Patrimônio Genético e do Conhecimento Tradicional Associado (SisGen n° A09F694).

The COP was characterized and applied in the development of nanoemulgels (NCOPs), which were then evaluated in terms of the preliminary safety profile and the in vivo antiedematogenic potential, as well as for the technological characterization and stability testing.

### 2.3. Chromatographic Conditions

#### 2.3.1. GC-MS for Chemical Composition Analysis of the *C. multijuga* Oleoresin

A gas chromatograph system (Shimadzu QP5000, Kyoto, Japan) coupled with a mass spectrometry (MS) detector was used to analyze the chemical composition of the crude COP and the standards of CAR and OX. For the separation of volatile compounds, a Durabond-DB5 capillary column (30 m × 0.25 mm × 0.25 µm) of fused silica was used. The temperature of the injector and ion source temperature were set at 220 °C and 250 °C, respectively. The oven temperature was programmed at 60 °C for 3 min and from 60 °C to 300 °C at a rate of 3 °C/min, finalizing a run time of 75 min, with helium as a carrier gas at a flow rate of 1 mL/min.

#### 2.3.2. GC-FID for Quantification of CAR and OX

For the quantification of CAR and OX contents in nanoemulgels and in crude COP, a previously validated method was applied [20]. The signal was recorded and processed with GC-Solution Data Analysis Software (Shimadzu).

#### 2.3.3. GC-MS for Skin Permeation/Retention Studies

To quantify the amount of CAR and OX permeated or retained in the skin layers, the chromatographic parameters validated by Lucca et al. (2015) were applied [21]. The signals were recorded and processed with GC-MS Solution Data Analysis Software (Shimadzu).

### 2.4. Chemical Characterization of C. multijuga Oleoresin

For chemical characterization, the crude COP (volatile fraction) was diluted in diethyl ether (2:100 *v*/*v*) before analysis, and a volume of 1.0 µL was injected (liquid injection) in split mode (1:40), according to the chromatographic parameters described in the Section 2.3.1. The volatile compounds were identified via the Mass Spectral Library NIST12 (National Institute of Standards and Technology, MD, USA). However, to confirm the identified compounds, the composition was also determined by retention index (RI) analysis [18]. The linear RI of each compound was calculated (Appendix A based on the retention time of the peaks and a homologous series of n-alkanes (C9–C30). Similarly, the CAR and OX standards were also analyzed under the same conditions to compare the chromatograms with the COP sample.

For the quantification of CAR and OX in the COP sample, the chromatographic parameters described in the GC-FID for the quantification of CAR and OX (Section 2.3.2) were applied. The standard curves of CAR (0.14 to 0.70 µg/mL; r = 0.9981; product number 75541 Sigma) and OX (0.05 to 0.50 µg/mL; r = 0.9931; product number 91034 Sigma) were prepared in ultrapure water from stock solutions (1000.00 µg/mL in methanol). The results were expressed as mg CAR/g and mg OX/g of oleoresin.

### 2.5. Development of Nanoemulgels

For comparative purposes, three nanoemulgel formulations for each group (CAR, OX, and COP) were prepared at equivalent concentrations to the major compounds of COP. These formulations were denominated as NCAR (1, 2, and 3), NOX (1, 2, and 3), and NCOP (1, 2, and 3).

CAR, OX, and COP were carried in the oily core with MCT and Span^®^ 80, and an aqueous phase composed of a fixed proportion of Tween^®^ 20 and ultrapure water. In addition, as a vehicle control in the tests, a blank nanoemulgel (BN) was prepared, containing only Span 80 and MCT in the oily core. First, the nanoemulsions (o/w) of CAR, OX, COP, and blank were prepared according to the method described by Dias et al. (2014) in a high-pressure homogenizer [22]. For this, the oily and aqueous phases were mixed separately for 5 min with magnetic stirrers. Afterwards, the aqueous phase was poured into the oily phase under stirring. To reduce the droplet size, the formulations were agitated in an Ultra Turrax disperser (IKA^®^-Werke GmbH & Co. KG, Staufen, Germany) for 1 min at 9500 rpm and then in a high-pressure homogenizer (EmulsiFlex-C3, Avestin, ON, Canada) for six cycles at 750 rpm. Then, the topical nanoemulgels were formulated by adding HEC (1.0% *w*/*w*) to the nanoemulsions and left to swell overnight.

### 2.6. Physicochemical Characterization of Nanoemulgels

#### 2.6.1. Organoleptic Characteristics and pH Evaluation

All nanoemulgels were evaluated for the organoleptic characteristics of color, appearance, and odor. The pH was measured in a pH meter with an electrode for semi-solid formulations (model DM-20, Digimed, SP, Brazil).

#### 2.6.2. Determination of β-Caryophyllene and Caryophyllene Oxide Contents

The CAR and OX contents were determined by gas chromatography coupled with a flame ionization detector (GC-FID) using a previously validated methodology [20] and according to chromatographic parameters described in the Section 2.3.2. The standard curves of CAR and OX were prepared under the same conditions, and the results were expressed as relative percentages (±SD).

#### 2.6.3. Droplet Size, Polydispersity Index, and Zeta Potential

The droplet size (nm) and polydispersity index (PDI) were measured on a Zetasizer model Nanoseries ZN90 (Malvern Instruments, Westborough, MA, USA) by the dynamic light scattering technique, with samples previously diluted in ultrapure water (1:1000). The zeta potentials (mV) were determined by electrophoretic light scattering in the same equipment after diluting samples in a 1 mM NaCl aqueous solution (1:1000). All parameters were measured immediately after dilution.

#### 2.6.4. Rheological Behavior

The rheological profile was evaluated at 25 °C using a Brookfield Rotational Viscometer, model DV-II+ (Brookfield Engineering Laboratories, Middleboro, MA, USA), set at a rotational speed range of 0.0 to 12.0× *g* with spindle 21. The results are shown as shear stress (D/cm²) vs. shear rate (/s), and viscosity (cP) vs. shear rate (/s). To classify the rheological behavior, the mathematical models of Bingham, Ostwald (Power Law), Casson, and Herschel–Bulkley for fluids were applied.

### 2.7. Storage Stability Evaluation of Nanoemulgels

The nanoemulgels were packaged in amber glass bottles, stored at 4 °C, and protected from light. The variation of the organoleptic characteristics, CAR and OX contents, pH, droplet size, PDI, and ZP were measured following preparation (initial time) up to 30 and 60 days.

### 2.8. Bioadhesion Measurements

Bioadhesion measurements of the nanoemulgels were carried out using a Texture Analyzer (TA.XTplus Texture Analyzer; Stable Microsystem, Godalming, UK) equipped with a gel mucoadhesion probe (P/10, 10 mm diameter cylinder). To simulate topical application, porcine ear skin was used as a substrate. Fresh porcine ears were obtained from a local slaughterhouse (Ouro do Sul, Harmonia, RS, Brazil). The skin was removed from the outer part of the ear with a scalpel and cut into circle slices. The excess hair and fat were removed with scissors and the thickness was measured with a thickness gauge (Mitutoyo Corporation, Kawasaki, Japan). Prior to the experiments, the skin sections were rehydrated with PBS pH 7.4. Then, the porcine skin sections were carefully fixed on the probe, and 400 µL of each nanoemulgel (n = 6/group) was applied. Both the formulation and the skin were kept at 32 °C during the experiment to simulate the surface temperature of the human skin.

The maximum force (mN) to detach the skin from nanoemulgels, as well as the work of adhesion (mN.mm), were recorded and calculated in the Exponent software (version 6.1, Stable Microsystem, Godalming, UK).

### 2.9. In Vitro Skin Permeation/Retention Studies

The permeation/retention studies were performed using porcine ear skin (n = 6/group) in a Franz-type diffusion cell apparatus. The porcine ear was obtained and prepared according to the method described in the Section 2.8. For experiments, only skin cuts with a thickness of 0.90 to 1.10 mm were used [23]. The receptor fluid was prepared as described by Lucca et al. (2015) and maintained at 32 °C in diffusion cells [21]. A volume of 500 µL of nanoemulgels was placed on the skin. In order to evaluate whether the application of nanoemulgels favors skin permeation, a control group of crude COP, as well as CAR and OX standards at equivalent concentrations to NCOP-3, NCAR-3, and NOX-3 were used.

After 6 h of permeation, an aliquot of receptor fluid was collected. After this step, the stratum corneum (SC) was separated by the tape stripping technique, and the epidermis and dermis were separated with a scalpel and scissors. All layers were transferred to GC vials and 1000 µL of Milli-Q water was added. Chromatographic analysis was performed according to a previously validated method as described in the Section 2.3.3. For quantitative purposes, the CAR and OX standard curves were prepared both in skin matrix and receptor fluid. Thus, the results were expressed as µg/cm^2^ for skin samples and µg/mL for receptor fluid (±SD).

### 2.10. Hen’s Egg Chorioallantoic Membrane Test (HET-CAM)

The preliminary safety profile of developed nanoemulgels was evaluated by the HET-CAM assay. Fertilized chicken eggs were incubated under controlled temperature and humidity (37.5 ± 0.5 °C and 60.0–70.0% humidity) for 10 days. The analyzed groups (n = 6/group) were 0.9% *w*/*v* NaCl (negative control), 0.1 M NaOH (positive control of hemorrhage and coagulation), 1.0% *w*/*v* sodium lauryl sulfate (positive control of vasoconstriction), MCT (diluent of COP, CAR, and OX), BN, 10.0% CAR in MCT, NCARs, 5.6% OX in MCT, NOXs, 24.9% COP in MCT, and NCOPs.

A volume of 300 µL of each substance or nanoemulgel was applied to the chorioallantoic membrane (CAM) and the appearance of irritant effects—hemorrhage, coagulation, hyperemia, and vasoconstriction were observed over 5 min. [24,25]. For each sample, the irritation score was calculated according to Appendix A. The results were expressed as mean (±SD).

### 2.11. In Vivo Antiedematogenic Activity

#### 2.11.1. Animals and Ethics Statements

Male BALB/c mice (20 to 30 g, 4 weeks) were provided by IBEx (Inovação em Biotério de Experimentação, Universidade Feevale, Novo Hamburgo, RS, Brazil). All animals were maintained under standard conditions, 22 ± 1 °C at 50 ± 10% relative humidity, 12 h of the light-dark cycle with the lights on at 7:00 a.m., and with *ad libitum* access to water and food (Nuvital). This study was previously approved by the Animal Ethics and Research Committee of the Universidade Federal do Rio Grande do Sul (CEUA—UFRGS, protocol No: 36210). Experimental procedures were carried out in accordance with all recommendations for animal experimentation of the Conselho Nacional de Experimentação Animal of Brazil, National Institutes of Health Guide for the Care and Use of Laboratory Animals, and comply with the ARRIVE guidelines [26,27,28].

#### 2.11.2. Arachidonic Acid-Induced Mouse Ear Edema

To evaluate the in vivo antiedematogenic activity, the arachidonic, acid-induced mouse ear edema model was applied [29,30]. The animals (n = 6) were treated with 0.04 g of NCARs, NOXs, or NCOPs. To verify the effect of the nanoemulsified systems, crude COP, CAR, or OX standards (without the addition of diluents or permeation enhancers) were applied at a concentration equivalent to the highest nanoemulgel concentration. Blank nanoemulgel (BN) was also investigated to verify the absence of biological effects of the carrier system. As a positive control, a solution of indomethacin (3 mg) in acetone was applied. Meanwhile, acetone was used as a negative control. The concentration of each treatment is given in the Appendix A.

All treatments and control substances were applied to the posterior and anterior parts of the right ear. After 1 h of the application, the edema was induced in the right ear by a topical application of 20 µL of arachidonic acid solution in acetone (2 mg/ear). On the left ear of each animal, 20 µL of acetone was applied. After 1 h of induction, the animals were euthanized by an intraperitoneal injection of ketamine (100 mg/kg) together with xylazine (10 mg/kg). Then, 6 mm^2^ samples of both ears were collected and weighed. Ear edema inhibition (%) was measured according to Appendix A, by the weight difference between the right and left ears of each animal. For subsequent quantification of myeloperoxidase (MPO), samples were stored at −80 °C.

#### 2.11.3. Histological Microscopy Study

For histological examination, mouse ear tissues were fixed in a solution of 10% *v*/*v* formaldehyde in PBS at pH 7.2. Afterwards, the tissues were dehydrated in ethyl alcohol solutions at an increasing concentration (70.0 to 100.0% *v*/*v*). After inclusion in Paraplast, cross sections were formed in a microtome and stained with hematoxylin-eosin.

#### 2.11.4. Myeloperoxidase Assay

The enzyme MPO was measured using the spectrophotometric method described by De Young (1989) [31]. The absorbances were measured at 630 nm (Spectramax M5, Molecular Devices, San Jose, CA, USA) and the results were expressed as optical density/biopsy (±SD).

### 2.12. Statistical Analyses

Statistical analyses were performed by GraphPad Prism software (version 6.0), with Student’s *t*-test (unpaired *t*-test) and one-way ANOVA with a post hoc Tukey test. The correlation between the amount of CAR and OX permeated in the epidermis and dermis with the in vivo edema inhibition (%) was determined by Pearson’s correlation coefficient (r > 0.99). The significance level was set at 5.00%.

## 3. Results

### 3.1. Chemical Composition of Copaifera multijuga and Its Major Compounds

The total ion chromatograms of COP and the standards (CAR and OX) are shown in Appendix A. Nineteen compounds were identified in the COP sample, with CAR (36.51%) and OX (21.57%) being present as the major compounds. The detailed composition of COP is described in Appendix A. The quantification of these compounds confirmed the similar proportionalities in COP, at 365.00 ± 12.00 mg CAR/g and 216.00 ± 8.05 mg OX/g.

### 3.2. Development, Characterization, Storage Stability, and Bioadhesion Behavior of Nanoemulgels

The development of the formulations was based on the optimization study of *Copaiba* oil-based nanoemulsions by Dias et al. (2014), in terms of the number of cycles and pressure of the high-pressure homogenizer [22]. However, in this study, the influence of the different proportions of the surfactants (1.0 to 3.0% *w*/*w*) in the oil and aqueous phases and the proportion of the gelling agent (1.0 to 3.0% *w*/*w*) were evaluated (data not shown), aiming to obtain stable systems with a higher proportion of oil core and droplet size inferior to 300 nm. After preliminary tests, nine nanoemulgels were developed: NCAR-1, NCAR-2, NCAR-3, NOX-1, NOX-2, NOX-3, NCOP-1, NCOP-2, and NCOP-3. The percentage composition of each formulation, as well as the concentrations of CAR and OX (mg/g) in each nanoemulgel, are given in Table 1. Additionally, a blank nanoemulgel (BN) was prepared for comparative purposes.

All formulations had a homogeneous appearance, were white or had a slightly yellowish color, and had a characteristic odor of CAR (woody-spicy) or OX (woody-mint). The results of the nanoemulgel characterizations throughout the storage of up to 60 days are shown in Appendix A. All developed nanoemulgels, were classified as monodisperse nanometric systems (droplet size < 500 nm; PDI < 0.3), with a low tendency for droplet aggregation (PZ < −30 mV), according to droplet size, PDI, and ZP values reported by others’ studies [14,16,22,29,32].

With regard to the contents of CAR and OX in the nanoemulgels, all formulations presented a content superior to 96%, relative to the initial value incorporated, indicating that the nanoemulsified system ensures stability by reducing content loss through volatilization. The average pH value was 6.0 with minimal variations (Appendix A), which is considered acceptable for topical application (ideal pH 5.5–6.5) [33].

With the topical application route in mind, formulations that exhibit a pseudoplastic rheological profile are desirable, as this flow characteristic is associated with ease of application and patient acceptability [34,35]. The rheograms of the nanoemulgels are shown in Appendix A. All formulations presented a decrease in viscosity (cP) in relation to the increase in shear rate (D/cm^2^), thus characterizing the nanoemulgels as non-Newtonian fluids (Appendix A). In the shear stress vs. shear rate graphs (Appendix A), the curve intersection at the origin is indicative of pseudoplastic behavior. Therefore, the formulations have no thixotropic effect [34]. By applying the mathematical models, a better correlation coefficient was obtained by the Ostwald equation, with r > 0.999 for all nanoemulgels, thus confirming the rheological behavior of the nanoemulgels having non-Newtonian and time-independent fluid behavior [35].

As the nanoemulsified systems have the same proportion of oily core without variations in the proportion of surfactants and polymeric agent, as well as the same overall charge, the bioadhesion was similar between them and thus showed no significant variation in the residence time on the skin. According to the results (Figure 2), the formulations did not differ statistically in terms of detach force (mN) and work (mN.mm).

### 3.3. In Vitro Skin Permeation/Retention Studies

The results of the in vitro skin permeation/retention study are shown in Figure 3, and the mass spectra of both substances, CAR and OX, after the skin sample extraction process and subsequent analysis using GC-MS, are shown in Appendix A. The incorporation of CAR, OX, and COP in the nanoemulsified systems enabled a considerable increase in the permeation of CAR and OX in the deeper layers of the skin when compared to the control groups of COP, CAR, and OX alone. The amount of CAR and OX retained in the epidermis was proportional to the concentration of these substances in the oily core, i.e., low, medium, and high concentrations, within each analyzed group (NCARs, NOXs, and NCOPs). On the other hand, this proportionality was not evident in the dermis for NCOP and NOX, between the medium and high concentrations, suggesting a possible saturation in this layer (NCOP-2 and NCOP-3, *p* > 0.999—Figure 3A; NOX-2 and NOX-3, *p* = 0.2180—Figure 3B).

Moreover, among the in vitro/ex vivo models applied for skin permeation evaluation, the in vitro model with ear porcine skin using a Franz diffusion cells apparatus is considered the gold standard. The porcine ear skin shows physiological similarity to human skin, especially in the thickness of the epidermis and lipid composition, thus conferring permeability very similar to that of human skin [36,37,38]. Several studies have compared the in vitro skin permeability of different compounds in animal and human skins and have found excellent correlation indices between porcine skin and human skin, especially for lipophilic substances [39,40]. Additionally, thus, the in vitro model with porcine skin can predict in vivo skin permeation [36,41]. Furthermore, when compared to other animal skin and human skin, porcine skin has the advantage of availability/ease of acquisition.

### 3.4. Hen’s Egg Chorioallantoic Membrane Test (HET-CAM)

Before in vivo assay, the potential irritating effects of the nanoemulgels were measured using the alternative method of HET-CAM [24,25]. The formulations and control groups were applied to the chorioallantoic membrane (CAM) of fertilized chicken eggs, the results of which are presented in Figure 4. As illustrated, the effects of hemorrhage and coagulation were observed only in the positive control (A—0.1 M NaOH), with an irritation score (IS) of 13.35 ± 0.22 conferring a severe irritant (IS > 9.0). The vasoconstriction effects were observed only after the application of 1% *w*/*v* sodium lauryl sulfate, with an IS of 10.62 ± 0.22, also classified as a severe irritant. The other groups showed no irritating effects on the CAM. All nanoemulgels, as well as the respective controls of CAR, OX, and COP in MCT, did not differ from the negative control of 0.9% *w*/*v* NaCl (*p* > 0.05).

### 3.5. In Vivo Antiedematogenic Activity and Correlation with Skin Permeation Profile

After ensuring the safety of the formulations, the in vivo antiedematogenic activity was evaluated using an arachidonic, acid-induced edema model in a mouse ear. The results are shown in Figure 5, and supplementary histological sections are given in Appendix A. Regarding the mass of edema (Figure 5A), ears treated with NCOP-2 and NCOP-3 did not differ statistically from the positive control, indomethacin, suggesting similar antiedematogenic activity. It is important to highlight that the nanocarrier system alone, evaluated by the application of BN, did not differ from the negative control, acetone, showing no antiedematogenic activity.

Increasing concentrations of CAR, OX, and COP in the oily core of the nanoemulgels caused a reduction in edema mass and, consequently, an enhancement of edema inhibition. The quantification of MPO (Figure 5B) revealed low values of optical density/biopsy for all groups, associated with the low migration of neutrophils to the site of inflammation at the time evaluated [30,42,43]. Histological images (Appendix A) also did not show the significant migration of PMNCs but provided evidence of vasodilation, as vascular permeability could be observed.

The conversion of the edema mass values into the percentage of edema inhibition can be seen in Figure 5C,D. NCOP showed a statistical difference in antiedematogenic activity compared to NCAR for medium and high concentrations. In Figure 5(c1,c2), the correlations between the edema inhibition and the amount of CAR permeated in the epidermis and dermis for NCAR and NCOP, respectively, are presented. Both NCAR and NCOP have a linear correlation in the epidermis, whereas a non-linear correlation was observed in the dermis for the NCOP samples. In a complementary way, in Figure 3A, a saturation of CAR in the dermis released from NCOP-2 and NCOP-3 was observed, which is reflected in the non-linear correlation (Figure 5(c2)). Similarly, a linear correlation is observed for the amounts of OX permeated in the epidermis from NOX and NCOP (Figure 5(d1,d2)). A possible saturation of OX in the dermis was also evidenced by a non-linear correlation (r = 0.7885) from the application of NOXs (Figure 3B and Figure 5(d1)).

There was no statistical difference between NOXs and NCOPs in terms of edema inhibition (Figure 5D), which could suggest that OX is the main compound responsible for the antiedematogenic action of the *C. multijuga* oleoresin. However, the permeation data (Figure 3B) showed a 40-fold increase in the permeation of OX from NOXs compared to NCOPs. Thus, for the edema-inhibiting effect to occur (Figure 5D), OX must be at a much higher concentration in the epidermis and dermis.

## 4. Discussion

Despite the pharmacological potential reported in the literature for fatty acids obtained from *Copaifera* oleoresins [28,29], this study focused on the characterization of the volatile fraction since this is where the CAR and OX are present. From the quantification of the content of these compounds in the COP, it was possible to develop nanoemulgels with equivalent concentrations to conduct an accurate comparative study.

The composition of COP determined in this study was in accordance with previous studies, which reported contents of CAR in the range of 5.08 to 62.70% and OX from 0.22 to 58.40% [9,10,16,30]. Together, these data support the definition of CAR and OX as major sesquiterpenes of *C. multijuga* oleoresin. CAR and OX, as isolated compounds, as well as the oleoresin itself, have an unctuous appearance and volatility instability. Based on these undesirable characteristics, nanoemulgels were developed for each group under evaluation.

After the characterization of the COP chemical composition by GC-MS and quantification by GC-FID, three nanoemulsions of each group (CAR, OX, and COP) were prepared at equivalent concentrations to the active substances quantified in COP. The nanoemulsion oily cores were standardized at 33% *w*/*w* and the gelling agent at 1% *w*/*w*, since different proportions of oily core and the gelling agent could lead to variable permeation profiles. In relation to the rheological profiles, the small differences among NCAR, NOX, and NCOP could be due to the specific densities of the bioactive compounds and oleoresin in the oily core. Thus, it can be concluded that the viscosity is dependent on the characteristics of the oily core since all formulations have similar droplet sizes and the same amount of gelling agent (1% *w*/*w* HEC).

To confirm the similarity of the formulations for topical application, a complementary bioadhesion test was performed. Bioadhesion can be defined as the ability of two materials to be held together by interfacial forces, in which at least one of the materials corresponds to a biological matrix [44]. Experimentally, this property depends mainly on the composition of the formulation, considering that the different compounds have different interfacial forces and viscosity [45]. Therefore, the small variations observed in the rheological profiles do not result in differences in the bioadhesion.

In relation to permeation profiles, CAR, in equivalent concentrations in the oily core, showed greater permeation capacity when associated with the other components of oleoresin in NCOPs. It is important to note that the higher the concentration of COP in the oily core, the lower the concentration of MCT. Although MCT is considered a skin permeation enhancer [15,46], NCOP-3, with 5.1% *w*/*w* MCT (lowest concentration), showed a greater amount of CAR permeation between the skin layers. This supports the hypothesis that the permeation of CAR is facilitated by the presence of the other oleoresin components.

In contrast, an inverse relationship was observed for the cutaneous permeation of OX, which permeated less when associated with the oleoresin. The amount of OX permeated in the epidermis was 40-fold higher from NOX-3 than from NCOP-3. When compared to the amount permeated from the controls of OX and COP not carried in nanoemulgels, it was also possible to observe a difference of up to seven-fold higher in the epidermis, reinforcing the assumption that OX permeation is facilitated when not associated with the oleoresin.

There are many factors that influence the permeation process of nanodroplets through the skin layers. Among them, those associated with the formulation characteristics are droplet size, surface charge, shape, and chemical composition [14,46]. The nanoemulgels had a similar droplet size, and the NCOP-3, despite having a 100 nm increase, compared to the other formulations, did not have an influence on its permeation. In addition, the nanoemulsions were obtained by the same method and under the same conditions resulting in droplets with the same shape and similar ZP. Therefore, the only factor that may have influenced the differences in the permeation profile was the chemical composition, with respect to the presence of CAR, OX, and COP in the oily core. Factors such as molecular weight and lipophilia (Log*P*) of these compounds could also influence their permeation [15,46].

CAR has a molecular weight of 204.35 g/mol and a Log*P* of 4.4, whereas OX is 220.35 g/mol and has a Log*P* of 3.6 [47,48]. The ability of these two compounds to interact with phospholipid bilayers was investigated in an in vitro model and by differential scanning calorimetry. The main results showed that both compounds interacted with the membrane model, and that CAR had a greater fluidizing effect than OX [49]. In addition, based on the hydrophobic character of the SC and Log*P* value, CAR is slightly more lipophilic than OX, which may have facilitated its passage through the SC barrier.

The application of the HET-CAM assay attested to the safety profile of the formulations prior to in vivo application. By applying the arachidonic, acid-induced mouse ear edema model, the evaluated nanoemulgel groups demonstrated anti-inflammatory potential. Bearing in mind the mechanisms of action already explored in the scientific literature, we suggest that the antiedematogenic and anti-inflammatory action of CAR and OX occur by inactivating COX-2, decreasing the migration of PMNCs, and stabilizing free radicals generated during the inflammatory process. In this sense, the reduction in inflammatory effects, such as vasodilation, erythema, and pain, can be associated with a reduced release of inflammatory mediators by the action of CAR and OX [2,10,50,51].

Furthermore, when it comes to the topical application of CAR, it is important to discuss its action in the endocannabinoid system. CB2 receptors are present in peripheral tissues and play a vital role in maintaining the cutaneous immunity associated with the physical barrier of the SC and other dermal immune cells. In a molecular docking study, CAR showed a strong affinity for the CB2 receptor as a functional agonist [10,52]; therefore, its analgesic and anti-inflammatory actions could be explained by its binding to CB2 receptors on sensory nerves, keratinocytes, and immune cells. Keratinocytes, in the presence of skin damage, release pro-inflammatory cytokines, and CAR acts by modulating this release and later inhibiting the formation of edema [10]. In addition, different studies that have investigated the action of CAR in in vivo inflammatory models, such as vascular inflammation [53], colitis [11], neuroinflammation (Alzheimer-like phenotype model) [54], and arthritis [4], associated the anti-inflammatory effects of this compound through a cross-activation between the CB2 receptor and the peroxisome proliferator-activated gamma receptor (PPRA-γ). These studies suggested that the binding of CAR to the CB2 receptor results in intracellular signaling is capable of increasing the expression of PPRA-γ and consequently stimulating the release of anti-inflammatory cytokines, as well as reducing pain. Thus, this could explain the linear correlation found in this study between CAR permeation in the epidermis and dermis with edema inhibition.

On the other hand, OX, despite being an oxidized derivative of CAR, has no affinity for CB2 receptors. Its mechanism of action is not yet fully understood, but this molecule also has analgesic, antioxidant, and anti-inflammatory properties [10,55]. If the permeated amounts of CAR and OX (Figure 3) and similar percentages of inhibition are considered (Figure 5), OX must be in much higher concentrations than CAR to have antiedematogenic action. Furthermore, to obtain similar edema inhibition percentages, the range of CAR that permeated varied from 6.80 to 60.00 µg/cm^2^, and from 123.80 to 1128.20 µg/cm^2^ for OX, thus suggesting a dose-dependent relationship and greater potency for CAR due to its multi-target action.

In order to better understand how the inhibition of edema was influenced by the in vitro permeation data and the characteristics of the formulations, regarding droplet size, bioadhesion force, and viscosity, the data were plotted in a radar graph (Appendix A). From the traced profiles it can be seen that the formulations have overlaps (Appendix A) in the droplet size, viscosity, and bioadhesion force variables, so this suggests that these parameters do not contribute to differences in biological activity. Regarding the permeation of CAR and OX in the epidermis and dermis, the formulations differed, implying an influence of these parameters. In Appendix A, it can be seen that the equivalent formulations had similar profiles, and the inhibition of edema has a direct relationship with the CAR amount permeated.

The formulations of NCOP and NOX also differed in terms of the permeation of OX in the epidermis and dermis (Appendix A). For both groups of formulations, the edema inhibition increases with the amount of OX permeated. However, as previously discussed, the amount of OX that permeated from NCOPs was much lower than that of NOXs. In this way, for NCOP the antiedematogenic activity seems to be dependent mainly on CAR permeation.

## 5. Conclusions

The developed nanoemulgels did not differ significantly in terms of the physicochemical parameters evaluated, indicating that the differences observed in the in vitro permeation profiles and in vivo antiedematogenic activity (ear edema model) among the groups were related to the concentrations of the sesquiterpenes in the oily core. When comparing the cutaneous permeation profile of the two sesquiterpenes, obtained by the in vitro model with porcine ear skin disposed in Franz diffusion cells, it was possible to observe greater amounts of caryophyllene oxide retained in the epidermis and dermis compared to the amounts of beta-caryophyllene. Thus, in relation to beta-caryophyllene, caryophyllene oxide can be considered a compound with lower potency since it must permeate in an amount 20-fold greater than beta-caryophyllene to perform similar antiedematogenic activity. On the other hand, it is interesting to observe that the permeation of caryophyllene oxide, when isolated in the nanoemulgel, is on average 40-fold higher than when mixed with other constituents of *C. multijuga* nanoemulgel. The opposite behavior is observed for beta-caryophyllene, which permeates on average 1.5-fold more when mixed with other constituents of *C. multijuga* nanoemulgel than when it is isolated in the nanoemulgel. New perspectives arise from the findings in this study, the main ones being the understanding of the dose–response relationship of these compounds and the permeation process of the sesquiterpenes when simultaneously present. So far, these results contribute to the investment of nanostructured systems for the topical release of sesquiterpenes in order to improve their permeation profiles and potentiate their bioactive effects.

## Figures and Tables

**Figure 1 biomolecules-12-01102-f001:**
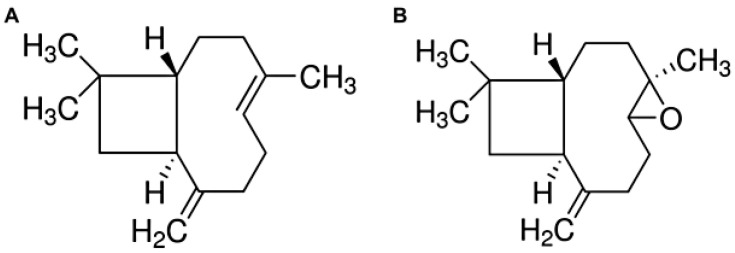
Chemical structures of β-caryophyllene (CAR) (**A**) and caryophyllene oxide (OX) (**B**).

**Figure 2 biomolecules-12-01102-f002:**
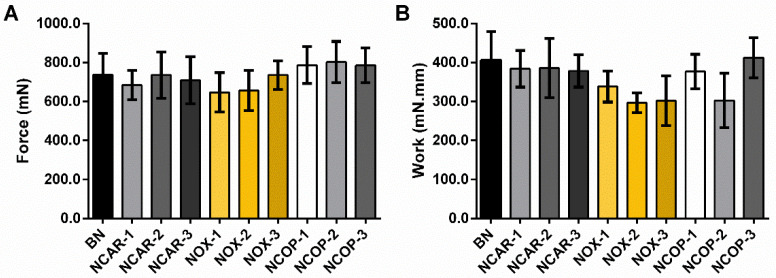
Bioadhesive parameters of nanoemulgels of β-caryophyllene (NCARs), nanoemulgels of caryophyllene oxide (NOXs), and nanoemulgels of *C. multijuga* oleoresin (NCOPs). Force (mN) (**A**); Work of adhesion (mN.mm) (**B**). The nanoemulgels (n = 6/group) were put in contact with the skin by a force of 2.5 N for 180 s. After, the probe was lifted at a constant rate of 1.0 mm/s. Data are reported as mean ± SD and were statistically evaluated using one-way ANOVA followed by Tukey’s post hoc test (*p* > 0.05).

**Figure 3 biomolecules-12-01102-f003:**
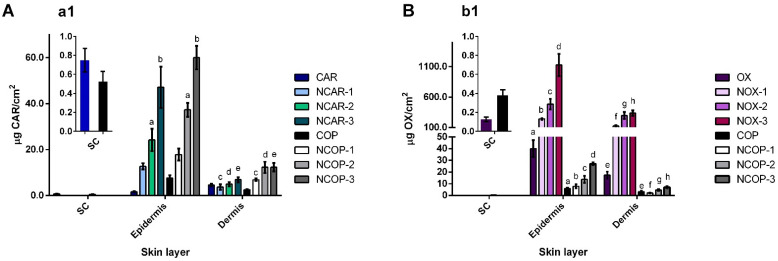
Cumulative amount permeated (µg/cm^2^) of β-caryophyllene (CAR) (**A**) and caryophyllene oxide (OX) (**B**) in skin layers with detailed permeation in the stratum corneum (**a1**,**b1**). The presence of CAR and OX was not detected in the samples of receptor fluid. Same letters mean the statistical difference between groups (n = 6/group) with proportional amounts of CAR and OX in the oily core, performed by one-way ANOVA followed by Tukey’s post hoc test (*p* < 0.001).

**Figure 4 biomolecules-12-01102-f004:**
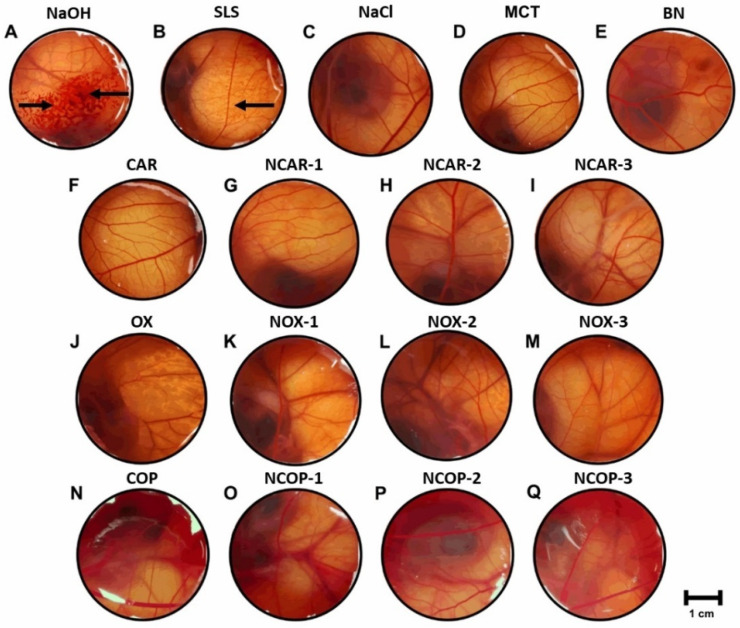
Images of HET-CAM assay after 5 min exposure (n = 6/group): (**A**) 0.1 M NaOH (IS 13.35 ± 0.22); (**B**) 1.0% *w*/*v* Sodium lauryl sulfate (IS 10.62 ± 0.22); (**C**) 0.9% *w*/*v* NaCl (IS 0.00); (**D**) MCT—Medium chain triglycerides (IS 0.00); (**E**) BN—Blank nanoemulgel (IS 0.00); (**F**) CAR—10.0% *w*/*w* β-caryophyllene in MCT (IS 0.00); (**G**) NCAR-1—nanoemulgel of β-caryophyllene (IS 0.00); (**H**) NCAR-2 (IS 0.00); (**I**) NCAR-3 (IS 0.00); (**J**) OX—5.6% *w*/*w* caryophyllene oxide in MCT (IS 0.00); (**K**) NOX-1—nanoemulgel of caryophyllene oxide (IS 0.00); (**L**) NOX-2 (IS 0.00); (**M**) NOX-3 (IS 0.00); (**N**) COP—24.9% *w*/*w C. multijuga* oleoresin in MCT (IS 0.00); (**O**) NCOP-1—nanoemulgel of C. multijuga oleoresin (IS 0.00); (**P**) NCOP-2 (IS 0.00); (**Q**) NCOP-3 (IS 0.00). Arrows indicate the effects of hemorrhage and coagulation (**A**), and vasoconstriction (**B**). Data were evaluated by one-way ANOVA followed by post hoc Tukey’s test—All groups evaluated (**C**–**Q**) differed in relation to control groups of hemorrhage (**A**) and vasoconstriction (**B**) (*p* < 0.01).

**Figure 5 biomolecules-12-01102-f005:**
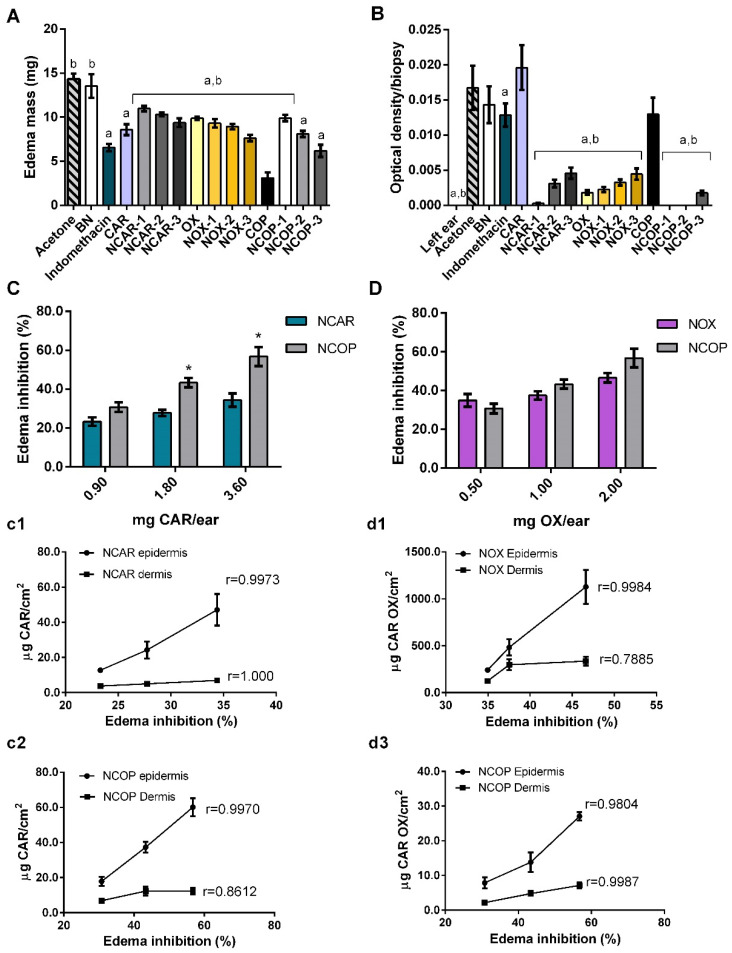
Results of in vivo antiedematogenic assay using arachidonic acid-induced edema in a mouse ear (n = 6/group). All data were expressed as mean ± SD. Results expressed as edema mass (mg) per group, n = 6 (**A**), and optical density/biopsy per group from MPO assay, n = 6 (**B**). a Statistical difference from negative control (Acetone), and b Statistical difference from positive control (Indomethacin), assessed by one-way ANOVA followed by Tukey’s test (*p* < 0.001). Edema inhibition (%) vs. mg of β-caryophyllene (CAR) topically applied on ears, blue bars corresponding to nanoemulgels of β-caryophyllene (NCARs) and gray bars to nanoemulgels of *C. multijuga* oleoresin (NCOPs) (**C**). Correlation curves between edema inhibition (%) and the cumulative amount of CAR permeated in the epidermis and dermis from NCAR (**c1**) and NCOP (**c2**). Edema inhibition (%) vs. mg of caryophyllene oxide (OX) topically applied on ears, purple bars corresponding to nanoemulgels of caryophyllene oxide (NOXs) and gray bars to NCOPs (**D**). Correlation curves between edema inhibition (%) and the cumulative amount of OX permeated in the epidermis and dermis from NOX (**d1**) and NCOP (**d2**). * Statistical difference between equivalent concentration groups by one-way ANOVA followed by Tukey’s test (*p* < 0.05).

**Table 1 biomolecules-12-01102-t001:** Percentage composition (% *w*/*w*) of COP, CAR, and OX nanoemulgels.

Phase	Component	BN	NCAR			NOX			NCOP		
			1	2	3	1	2	3	1	2	3
Oily core	MCT	30.0	27.5	25.0	20.0	28.6	27.2	24.4	23.8	17.6	5.1
	Span^®^ 80	3.0	3.0	3.0	3.0	3.0	3.0	3.0	3.0	3.0	3.0
	CAR	-	2.5	5.0	10.0	-	-	-	-	-	-
	OX	-	-	-	-	1.4	2.8	5.6	-	-	-
	COP	-	-	-	-	-	-	-	6.2	12.4	24.9
Aqueous phase	Tween^®^ 20	1.0	1.0	1.0	1.0	1.0	1.0	1.0	1.0	1.0	1.0
	Water	66.0	66.0	66.0	66.0	66.0	66.0	66.0	66.0	66.0	66.0
Gelling phase	HEC	1.0	1.0	1.0	1.0	1.0	1.0	1.0	1.0	1.0	1.0
Content	[ ] mg CAR/g	-	22.75	45.50	91.00	-	-	-	22.63	45.30	90.90
	[ ] mg OX/g	-	-	-	-	13.57	27.10	54.27	13.40	26.80	53.70

BN—blank nanoemulgel; NCAR—nanoemulgel of β-caryophyllene; NOX—nanoemulgel of caryophyllene oxide; NCOP—nanoemulgel of *C. multijuga* oleoresin; MCT—medium chain triglycerides; HEC—hydroxyethylcellulose. The numbers 1, 2, and 3 associated with the nanoemulgels represent low (1), medium (2), and high (3) concentrations in each group. [ ] concentrations corrected for degree of purity.

## Data Availability

Not applicable.

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
