# Peer review of "Correlation between the Skin Permeation Profile of the Synthetic Sesquiterpene Compounds, Beta-Caryophyllene and Caryophyllene Oxide, and the Antiedematogenic Activity by Topical Application of Nanoemulgels"

_biomolecules, 2022, doi:10.3390/biom12081102_

Round 1

Reviewer 1 Report

In the manuscript entitled "Correlation between the skin permeation profile of the sesquiterpene compounds, beta-caryophyllene and caryophyllene oxide, and the antiedematogenic activity by topical application of 4 nanoemulgels" authored by Patrícia Weimer  et al, the authors present in a clear and balanced manner the aim, methods and results. The selected protocol includes standardized methods and assays, the results are clearly described and discussed, still two recommendations are to be considered.

Point 1- modify the title to indicate the source of the sesquiterpene compounds  

Point 2 - point out the novelty of the study as part of the introduction

Point 3 - rephrase and/ or develop the conclusion section (abstract and text body) to underline the study's importance and novelty.

Author Response

We thank the Reviewer #1 for the attention given to the manuscript and his/her accurate comments.

Point 1- modify the title to indicate the source of the sesquiterpene compounds.

Special thanks for this comment; the title was modified as requested.

Point 2 - point out the novelty of the study as part of the introduction.

Modifications were made to the Introduction to highlight the novelty of this study.

Point 3 - rephrase and/ or develop the conclusion section (abstract and text body) to underline the study's importance and novelty.

Special thanks for this comment. All corrections were made in the conclusion and abstract sections as requested.

Reviewer 2 Report

The review is attached.

Author Response

We thank the Reviewer #2 for the attention given to the manuscript and his/her accurate comments.

  1. Linguistic and punctuation errors occur, including:
  2. on line 17, instead of "in", there should be "on" in the phrase "influence on nanostructured system";
  3. on line 29 should be used the plural in "amount";
  4. on line 31, instead of "association in oleoresin", there should be "association with oleoresin";
  5. on line 51, instead of "interaction with on cannabinoid 2 (CB2) receptors", it should be "interaction with cannabinoid 2 (CB2) receptors";
  6. on lines 78-79 sentence should be changed to "Technological characterization of the formulations and in vitro safety profile assessment were also conducted."

We apologize for our inattention; grammatical errors were corrected, and the manuscript was spell-checked by a professional editing service.

  1. In accordance with the journal's requirements, the company's headquarters producing the equipment or materials should be provided.

The required information was added to the text.

  1. The literature contains only six items in the last five years (13%), proving that it is not up-to-date.

We understand the reviewer's concern on this point; we have done our best to include updated references, and at the same time, we chose to keep references from the original studies.

Questions:

  1. Was the skin impedance tested before starting the measurement of permeation?

Special thanks for this comment. The skin impedance was not measured before conducting the permeation study; however, some precautions were taken prior to the experiments. Fresh porcine ears obtained from a local slaughterhouse (Ouro do Sul, Harmonia, RS, Brazil) were used for the permeation assays. For this, the skins were removed with the aid of a scalpel from the same tissue region (outer part of the porcine ear). Samples with hematomas or visible vascular changes were discarded. After removal of the skins, the thickness was standardized, according to the OECD guideline (OECD 428, 2004), which comprises the partial removal of subcutaneous fat to standardize skin thickness. Furthermore, the supplier of the skins was validated (our partner for over 10 years), in terms of animal care, standardization of the age and weight of the animals, minimizing the influence of these variables on the structure of the skin, as well as on its bioimpedance.

Reference:

OCDE 428, 2004. OECD - GUIDELINE FOR THE TESTING OF CHEMICALS: Skin Absorption: in vitro Method. Test 1–8. https://doi.org/10.1787/9789264071087-en

  1. Why have the permeation kinetics not been studied and the permeation profile and permeation parameters not determined

Special thanks for this comment. We understand that the presentation of permeation parameters and permeation kinetics could enrich our study. However, we chose not to present them based on our main objective, which focused on the correlation of permeation (ex vivo) with in vivo inflammatory model data. In this context, the data on percent edema inhibition and amounts of the compounds retained in the skin layers were adequate to answer the problem raised. Since the anti-inflammatory activity was assessed in endpoint mode (single analysis time) we chose not to detail the pharmacokinetic profile of the sesquiterpenes in this study.

Moreover, based on the data from this study and centered on the interest of our research group in evaluating the permeation profile of natural compounds, we are elaborating future studies that will comprise the in-situ evaluation of the permeation of terpenic compounds by the application of microdialysis techniques in rodents and open flow microperfusion (clinical study).

Reviewer 3 Report

1. The introduction is too general, it refers to the general action of even systemic individual components, and not enough overview is given about any form of these compounds for topical application. I think it is necessary to reformulate the introduction a bit.

2. Abbreviations are listed in the material and methods, we also have the listing of some excipients under the protected name somewhere under chemical. It is necessary to be uniform. For example, the chemical or pharmacopoeial name and the protected name of the manufacturer everywhere

3. 2.4. supplement data on standard and validation parameters of the given method.

4. 2.5. it remains unclear to the reader whether you took over the formulations of nanoemulsions, and therefore nanogels, or whether you just took them from the literature and incorporated active principles into them. In the introduction, objective, and up to this paragraph, it seems as if you were involved in formulation development, stability assessment... And here it seems to me that you took ready-made formulations of nanoemulsions and incorporated active principles into them

5. 2.5.3. specify the ratio in which the samples were diluted. Were the samples measured immediately after dilution, or was there some period after dilution until the sample was measured?

6. 2..6 why wasn't the test done at room temperature, under stress conditions?

7. The introduction and the goal are absolutely not in agreement with the other parts. It is completely correct that the research was done on experimental animals or parts of the skin of a certain animal... From the initial part, one gets the impression of this in vivo test, as if testing on humans. In the introduction, we should also mention these tests, why they are justified on these animals, and the results of these models compared to humans in previous studies...

8. Even in the results in Table 1, it is not clear whether you chose the oil phase/surfactant ratios during development... or you took them from the literature

9. The conclusion is very similar to the abstract, reformulate. Also, again, it is not clear from the conclusion what the tests were done on.

10. You don't have a single reference in the last three years, and generally, all references are very old.

Author Response

We thank the Reviewer #3 for the attention given to the manuscript and his/her accurate comments.

  1. The introduction is too general, it refers to the general action of even systemic individual components, and not enough overview is given about any form of these compounds for topical application. I think it is necessary to reformulate the introduction a bit.

Special thanks for this comment. Modifications were made to the Introduction to highlight the novelty of this study.

  1. Abbreviations are listed in the material and methods; we also have the listing of some excipients under the protected name somewhere under chemical. It is necessary to be uniform. For example, the chemical or pharmacopoeial name and the protected name of the manufacturer everywhere.

We apologize for our inattention. The chemical names have been specified in section 2.1 Chemicals and Reagents.

  1. 2.4. supplement data on standard and validation parameters of the given method.

Data about the standards has been added in Section 2.4 as requested. Regarding the quantification method applied, it was validated in the study by Dias et al. (2012) conducted by our research group. To evidence the applicability of this method for the oxygenated derivative of caryophyllene, we checked the linearity, precision, accuracy, selectivity, limits of quantification and detection and chromatographic parameters for the caryophyllene oxide molecule, the results being in accordance with the ICH guidelines for analytical method validation and our national resolutions (data not shown).

Reference:

Dias, D. de O.; Colombo, M.; Kelmann, R.G.; De Souza, T.P.; Bassani, V.L.; Teixeira, H.F.; Veiga, V.F.; Limberger, R.P.; Koester, L.S. Optimization of headspace solid-phase microextraction for analysis of β-caryophyllene in a nanoemulsion dosage form prepared with copaiba (Copaifera multijuga Hayne) oil. Anal. Chim. Acta 2012, 721, 79–84, doi:10.1016/j.aca.2012.01.055.

  1. 2.5. it remains unclear to the reader whether you took over the formulations of nanoemulsions, and therefore nanogels, or whether you just took them from the literature and incorporated active principles into them. In the introduction, objective, and up to this paragraph, it seems as if you were involved in formulation development, stability assessment... And here it seems to me that you took ready-made formulations of nanoemulsions and incorporated active principles into them

We apologize for the possible confusion; we tried our best to clarify the information in the manuscript. All formulations obtained in the study correspond to nanoemulgels, according to the methodology described in the 2.5 section and their compositions are described in Table 1. We would like to highlight that to obtain the nanoemulgels, we initially developed nanoemulsions by high pressure homogenization and then we added hydroxymethylcellulose (1% w/v). In this process, the high-pressure homogenizer parameters applied were based on the optimization study by Dias et al. (2014). In our study, we specifically evaluated the influence of different proportions of the surfactants (1.0 to 3.0% w/w) in the oily and aqueous phases and proportion of the gelling agent (1.0 to 3.0% w/w), aiming to obtain stable systems with higher proportion of oily core and droplet size less than 300 nm. To avoid any misunderstanding, we specified these details in the Section 3.2 Development, characterization, storage stability, and bioadhesion behavior of nanoemulgels.

Reference:

Dias D de O, Colombo M, Kelmann RG, Kaiser S, Lucca LG, Teixeira HF, Limberger RP, Veiga VF, Koester LS. Optimization of Copaiba oil-based nanoemulsions obtained by different preparation methods. Ind Crops Prod 2014; 59: 154–162

  1. 2.5.3. specify the ratio in which the samples were diluted. Were the samples measured immediately after dilution, or was there some period after dilution until the sample was measured?

The ratio of dilution was detailed in the text and all parameters were measured immediately after sample dilution.

  1. 2.6 why wasn't the test done at room temperature, under stress conditions?

We understand the reviewer's concern on this topic. The most comprehensive study carried out by our research group on the development of nanoemulsions covered the assessment of stability at 4 x 25°C and demonstrated the need for storage at 4°C to ensure long-term stability. Since the scope of the study was to focus on pharmacological effect associated with permeation, in this study, we only evaluated stability under refrigeration to guarantee the stability required. Future studies with selection of the most promising formulations will evaluate stability at different temperatures and larger scale production compared to this study.

  1. The introduction and the goal are absolutely not in agreement with the other parts. It is completely correct that the research was done on experimental animals or parts of the skin of a certain animal... From the initial part, one gets the impression of this in vivo test, as if testing on humans. In the introduction, we should also mention these tests, why they are justified on these animals, and the results of these models compared to humans in previous studies...

We apologize for the possible confusion; we tried our best to clarify the information in the manuscript. We added some details about the tests in the introduction section and included an explanation of the choice of porcine skin and rodent model in Section 3.3 In vitro skin permeation/retention studies. At the same time, we would like to point out that we have not mentioned possible clinical trials in the manuscript, only in vivo assay (animal), the ethical principles being in line and the project approved on this topic.

  1. Even in the results in Table 1, it is not clear whether you chose the oil phase/surfactant ratios during development... or you took them from the literature.

Similar to the answer provided in Section 4, we specify the details of the development of the formulations in Section 3.2 Development, characterization, storage stability, and bioadhesion behavior of nanoemulgels. However, surfactant ratios, in each phase, were evaluated in the range of 1.0 to 3.0% w/w (data not shown). The surfactant ratio of 3.0% w/w was chosen because it gave the highest stability and least variation in droplet size after addition of the gelling agent.

  1. The conclusion is very similar to the abstract, reformulate. Also, again, it is not clear from the conclusion what the tests were done on.

Special thanks for this comment. We tried our best to improve the conclusion section.

  1. You don't have a single reference in the last three years, and generally, all references are very old.

We understand the reviewer's concern on this point; we have done our best to include updated references, and at the same time, we chose to keep references from the original studies.

Round 2

Reviewer 3 Report

Accept in present form